# A Retrospective Analysis of Endovascular Stent Insertion for Malignant Superior Vena Cava Obstruction, Focusing on Anticoagulation Practices

**DOI:** 10.3390/curroncol32110601

**Published:** 2025-10-27

**Authors:** Joshua Walker, Amsajini Ravinthiranathan, Athanasios Diamantopoulos, Spyridon Gennatas, Alexandros Georgiou

**Affiliations:** 1Guy’s and St Thomas’ NHS Trust, Guy’s Hospital, Great Maze Pond, London SE1 9RT, UK; amsajiniravinthiranathan@qehkl.nhs.uk (A.R.); athanasios.diamantopoulos@nhs.net (A.D.); spyridon.gennatas@nhs.net (S.G.); alexandros.georgiou@gstt.nhs.uk (A.G.); 2School of Cancer & Pharmaceutical Sciences, King’s College London, Guy’s Campus, Great Maze Pond, London SE1 1UL, UK

**Keywords:** malignant superior vena cava obstruction, SVCO, superior vena cava syndrome, endovascular stent, venous thromboembolism, VTE, stent-thrombosis, anticoagulation, thrombosis, pulmonary embolism

## Abstract

Malignant obstruction of the superior vena cava—a crucial vein in the thoracic cavity—is often managed by inserting endovascular stents. While anticoagulation therapy is typically prescribed post procedure, optimal strategies remain undefined. Our retrospective analysis evaluated 49 cancer patients who underwent stenting for this condition. The majority presented with metastatic disease, with a median survival of 2.4 months following intervention. Despite anticoagulation therapy, a significant proportion developed thrombotic complications, either within the stent or systemically. Notably, treatment regimens were frequently modified after discharge, predominantly due to patient preference for oral rather than injectable medications. This evaluation highlights the substantial thrombotic risk these patients face despite therapeutic intervention. The findings underscore the necessity for coordinated multidisciplinary care that prioritizes individualized treatment plans aligned with patient preferences. Further prospective research is warranted to investigate the efficacy of direct oral anticoagulants and to develop tailored risk assessment tools for this specific population.

## 1. Introduction

Malignant superior vena cava obstruction (mSVCO) is an oncological emergency with significant implications for patient care and quality of life. Lung cancer accounts for 78–85% of presentations [1]. Over the last forty years, endovascular stenting has emerged as the first-line intervention for mSVCO in solid organ malignancy. To reduce the incidence of in-stent and systemic thromboembolism, systemic post-procedural anticoagulation therapy is frequently used, although a significant knowledge gap persists regarding the most effective strategy [2,3].

There is considerable variation in anticoagulation practices, with no consensus on optimal drug selection, appropriate dosing or duration of therapy [3,4]. Conventional approaches for treating cancer-associated thrombosis (CAT) include using low molecular weight heparin (LMWH) and more recently, direct oral anti-coagulants (DOACs) [5]. The SVC is unique in its nature, facing different conditions to most other veins vulnerable to clotting, as it channels blood at a higher speed using gravity [6]. It has therefore received special therapeutic consideration, and a broad range of anticoagulation strategies have been deployed following stent insertion including anti-platelet agents, vitamin K antagonists and heparins.

Clinical guidelines lack specific drug recommendations [7,8]. They are predominantly supported by small retrospective cohort studies and case series with a focus on stent-related outcomes [3,4,9]. The Cardiovascular and Interventional Society of Europe (CIRSE) guidelines for pre-procedural anticoagulation provide guidance including tools to stratify bleeding and clotting risk and indications for specialist hematology advice, but no specific drug recommendations [2].

Authors of larger observational studies have questioned the benefit of high dose systemic anticoagulation [6,10]. They cite lack of mortality benefit and bleeding rates as a reason to avoid long-term therapeutically dosed anticoagulation. Venous thromboembolism (VTE), however, remains a significant concern to those living with cancer. It is the second leading cause of death among cancer patients. Effective treatment can reduce mortality, symptomatic burden and rates of hospitalization [11,12].

Overall, the current literature relating to the management of mSVCO after SVC stenting fails to capture and address the interconnected interventional, oncological and hematological factors that govern decision-making. This real-world single specialist center retrospective analysis aims to address this gap and provide a benchmark for current practice that will enable further research and service improvement in this field.

## 2. Materials and Methods

### 2.1. Context

The evaluation was performed at a tertiary center in South-East London. This includes a major cancer-care hub providing oncological care to millions of people. The Interventional Radiology (IR) team is one of the largest in London, involving 12 consultants. Anticoagulation decisions are typically initiated by the IR consultant, communicated with the oncological and medical teams treating inpatients and discussed on a case-by-case basis as required with the anticoagulation hematology consultant team.

### 2.2. Sample Collection

Using a code-based database query in the radiology information system, a consecutive series of patients who had received an endovascular stent for SVCO were identified. Patients were included in the analysis if they (1) had radiologically confirmed partial or complete SVCO and (2) underwent endovascular stenting between July 2016 and May 2022. They were excluded if the obstruction was identified as having a benign or hematological cause, or if post stenting they were discharged to another hospital with no further contact with services. As a result, all included patients were followed up until death.

### 2.3. Data Collection

A standardized data collection form was piloted and developed. Data was manually extracted by two researchers (JW/AR) and cross-validated where possible between multiple hospital IT systems. In three patients, the findings were cross-checked between researchers to reduce inter-observer variability. Date of initial cancer diagnosis, stent insertion and death were determined by examining available multi-disciplinary, inpatient, outpatient and general practice (GP) records to improve accuracy and reliability. Demographic details, oncological disease status and details of prior and current treatments were recorded. Anticoagulation practice was tracked across inpatient and outpatient computer systems and cross-referenced with inpatient drug charts. We tracked initial anticoagulation regimen, reasons for regimen changes and the rate of specialist consultations. This included criteria for dose escalation, dose reduction and agent switching. Complication data was collected using radiology reports and notes-based systems.

### 2.4. Statistical Analysis

Data were analyzed using SPSS version 29.0.0.0. Numerical variables were presented in medians and interquartile ranges (IQRs), while the categorical variables were presented in absolute values and percentages. Analysis included generating Kaplan–Meier curves and a descriptive analysis of subgroups. To better interpret missing data, the characteristics and outcomes of patients for missing data categories were compared to the general cohort.

### 2.5. Ethical Considerations

The project was registered and approved with Guy’s and St Thomas’ hospital audit department (Audit number: 17761) and the team adhered to local institutional data protection guidelines.

## 3. Results

Initially, 72 cases were identified. There were 23 exclusions (12 due to benign cause, 8 due to hematological disease, 2 were duplicate records and 1 due to discharge post procedure to another hospital without any further information in our electronic notes), leaving 49 cases for analysis. Demographic details, malignancy classification, extent of metastatic disease and performance status at the time of diagnosis and stent insertion are shown in Table 1. Treatment details, complication and survival data of all patients is found in Appendix A.

### 3.1. Procedural Details

All patients reported symptoms of SVCO at the time of stent insertion. The most commonly presenting symptoms were facial/arm edema (*n* = 31, 63%); dyspnea (*n* = 25, 51%) and distended neck veins (*n* = 13, 27%). The median time from initial cancer diagnosis to stent insertion was 265 days (range 3–1906 days, IQR 42–704).

Both covered and uncovered stents were used: 26/49 (54%) cases involved a single uncovered stent and ten (20%) involved a single covered (nitinol) stent. Thirteen (26%) involved two stents deployed: with a similar proportion of patients receiving two covered, two uncovered or mixed stent type. Analyzing cases consecutively, there was a trend over time towards covered stents—3/25 (12%) in the first half and 15/24 (63%) in the second used at least one covered stent. In all cases, stents were inserted with the intraluminal administration of 5000 units of Heparin. Eighty per cent (42/49) of patients received high dose steroids peri-procedurally.

### 3.2. Procedural Outcomes

Initial radiological resolution of SVC obstruction was achieved in 48 (98%) cases. Twenty-four (49%) patients had a documented fast symptomatic benefit, with 5 (10%) reporting no improvement and 20 (41%) having no clear record.

### 3.3. Anticoagulation Practices

Before stent insertion, 21 (43%) patients were taking therapeutic anticoagulation.

The plan devised by the treating IR team was communicated to the responsible medical/oncological teams in a ‘procedure note’ in 41 (84%) cases. There was no documented plan in eight (16%) cases. In 7/41 (14%) cases, the IR plan advised a referral for a hematology opinion.

Forty-five (92%) patients were discharged from hospital following hospital admission that included stent placement. At the time of discharge, therapeutically dosed dalteparin (standard of 200 units/kg, adjusting for weight and renal function) was advised in 28/45 (62%) patients. The multiple anticoagulation regimens advised at discharge from hospital are shown in Table 2.

In the 45 (92%) patients that were discharged from hospital, there were 22 instances of therapy modifications following the initial plan provided by the IR team. Out of these 22 instances, 13 (59%) involved hematology consultation in the outpatient setting. The primary reasons for changes were 9 (41%) occasions of a change in agent due to patient preference, six occasions of a change in agent or dose due to thrombus detection and six occasions of changes to direct oral anticoagulants. Other factors were impaired absorption = 1 and bleeding = 1.

### 3.4. Complications

Major events included one patient who became acutely unwell intra-operatively with evidence of cardiogenic shock, thought to be due to pulmonary artery compression. This led to a critical care admission and death three days later. Three further patients died within a week, two with heavily pre-treated and widely metastasized cancer, performance status of 3 and unknown symptomatic benefit of stent insertion, one patient with treatment-naïve metastatic lung cancer (T4M3M1c), performance status of 3 and recorded immediate symptomatic benefit. For all patients, no single factor was identified as the cause of deterioration and death. Cause of death was identified as metastatic breast, esophageal and lung cancer, respectively.

Following stent insertion 27/49 (55%) patients underwent CT or ultrasound imaging, indicated by routine disease surveillance or new symptoms. These showed seven cases of stent thrombosis, four of stent stenosis/occlusion and one case of SVCO recurrence with progressive disease. In addition, there were ten additional thromboembolic events, including five pulmonary embolisms, one internal jugular thrombus, two subclavian thromboses and two lower limb deep vein thromboses. Two out of four patients discharged without anticoagulation were admitted shortly after with shortness of breath and later diagnosed with pulmonary embolism. Therapeutic anticoagulation was promptly begun. The other two patients did not have interval imaging before deterioration.

Nine out of the eleven (82%) patients found to have radiologically proven SVC thrombosis, stenosis or recurrence were categorized to performance status 0 or 1 at the time of stent insertion. For the other two patients, the performance status is unknown. Median survival for the group of seven patients with stent thrombosis was 147 days. Six out of seven (86%) cases of radiologically proven stent thromboses were prescribed a regimen of therapeutically dosed LMWH. There was one significant instance of bleeding noted, a case of hemorrhagic bursitis following a fall. This was managed conservatively by holding anticoagulation before eventual re-commencement.

### 3.5. Survival

Following stent insertion, 45 patients (92%) survived to discharge from hospital. From cancer diagnosis, median survival was 14.7 months (IQR: 6.2–28.2 months). One patient was not included in the analysis due to an unclear date of diagnosis. The Kaplan–Meier analysis is presented in Figure 1a.

From stent insertion, median survival was 2.4 months (IQR 1–4.9 months). All 49 patients were included in this analysis. The Kaplan–Meier analysis is presented in Figure 1b.

## 4. Discussion

### 4.1. Summary of Key Findings

In our study, endovascular stenting for mSVCO had a high rate of technical success, with 92% of patients surviving to be discharged from hospital. The majority of patients reported symptomatic benefit following stenting. There were also significant complications, including a high rate of venous thromboembolism, amidst a range of anticoagulation treatment strategies. Many patients demonstrated a preference for oral anticoagulation medications.

At the time of diagnosis, most patients had metastatic lung cancer, with 75% of all patients having at least one extra-thoracic metastasis and 25% of cases at least two. This poses a key consideration when determining thrombotic risk. Seventy-five percent of the patients in our cohort would therefore score at least 2 points on the ONKOTEV score, a recently validated VTE risk assessment model (RAM) scoring system for those with malignancy (scoring points for vascular compression and metastatic disease) [11,13]. The RAM has been effective at predicting VTE in hospitalized patients with lung cancer, although currently, no comparable system is available for patients with mSVCO. A score of 2 predicts a 12-month incidence of VTE of 19.4%. A score of > 2 would predict the incidence of VTE at 12 months to be 34%. Due to a lack of consistent data, the further two measures to complete the ONKOTEV score (Khorana score and previous VTE event) were not calculated; it is likely that more patients in our cohort would reach the 2 or >2 threshold.

Those with SVC thrombosis survived twice as long as the general cohort (median of 147 days compared to 72). This trend has led some authors to challenge the utility of long-term high dose anticoagulation [3,10]. It is important to note that 45% of our cohort were not investigated with CT or ultrasound imaging following stent insertion; therefore, the true rate of stent thrombosis or recurrence is unknown. Our analysis reveals the relative pre-stent fitness of those diagnosed with these complications. For example, 9/11 (82%) patients with stent thrombosis, recurrence or stenosis were identified to have a performance status of 0–1 following stent insertion, compared to 20% of the general cohort, revealing a possible investigation bias inherent of such retrospective analysis.

### 4.2. Comparisons with the Literature

Our median survival of 2.4 months following stent insertion is longer than the large retrospective cohort study by Ratzon et al. [10] (44 days, 1.44 months) and shorter than a systematic review by Aung et al. [3] (4.74 months). Notably, only 20% of the studies included in this review reported survival outcomes, indicating possible reporter bias. It is notable that 60% of our patients were treated between 2020 and 2022. The impact of COVID-19 pandemic on the investigation, management and survival of those with cancer and mSVCO remains to be fully understood.

The rate of complications in our cohort, including recurrence and stent thrombosis, was similar to some retrospective reviews, for example, Azizi et al. [4] found re-stenosis rates of 10.5% (95% CI 8.4–12.6%), but higher than others [3]. Discrepancies are probably explained by pooled data capturing different patient groups, including those with benign SVCO and patients with earlier stages of cancer and more controllable disease [14]. In contrast to previous studies, we present a wide variety of anticoagulation approaches. This reflects the real-world practice of prescribing decisions made by different specialists across two hospital sites with no definitive guideline recommendations on drug type and dose. Treatment groups were deemed too small to draw reliable associations between treatment, effect and side-effects while adjusting for confounders.

Three retrospective observational studies demonstrate notably low stent thrombosis rates, reporting rates of 7%, 6% and 1%, respectively [15,16,17]. All three studies used an antiplatelet agent with either heparin, coumadin or dalteparin. Fadgetet et al. [16] reported bleeding rates and found them to be significantly higher than our cohort at 8%, with one death due to concurrent thrombocytopenia. Khorana et al. [5] found up to 22% of cancer patients on injectable anticoagulation therapy preferred to change from injectable to oral agents.

### 4.3. Strengths and Weaknesses

To our knowledge, no previous publication has captured endovascular stenting with this broad focus, gathering oncological, hematological and IR data with prescribing decisions over time. Strengths of this project also include relative homogeneity of disease by using solid organ malignancy, predominantly lung. The near-complete consecutive series of patients had very low missing data regarding survival. Many key results, including complication data, anticoagulation prescriptions and desire to change to oral agent are comparable with larger studies, as described above, supporting the generalizability of the findings.

Weaknesses include the retrospective case note design, sample size and high missing data in domains of symptomatic benefit and performance status. The survival of patients with no performance status recorded was shorter than the average of our cohort. It is likely therefore that our data overestimates performance status. Our analysis is largely descriptive, as the many known confounding variables and small sample size of subgroups limits the conclusions about the impact of treatment decisions. The anticoagulation management plans at discharge reflect what was planned rather than what was truly administered. Patient persistence in taking injectable anticoagulation has been reported to be as low as 37% [18], and this might be anticipated to be lower as patients approach the end of their lives.

### 4.4. Implications for Clinicians/Policy Makers/Patients

Our data suggest the poor prognosis of many diagnosed with mSVCO due to solid organ malignancy and a dynamic range of prescribing decisions following stent insertion. The insertion of an endovascular stent should trigger a systematic and holistic assessment of patients that considers their key priorities and include discussing expected prognosis, details of anticoagulation therapy and a referral to palliative care when indicated.

There is little evidence supporting the mortality benefit of such therapies [19]. The clinical significance of VTE is also questionable given that endovascular stenting is used towards the end of life, with evidence suggesting that the rate of asymptomatic DVT in hospice populations (median survival of 44 days) can be as high as 34% [20]. Yet the high VTE burden in our cohort underlines why conventional wisdom prioritizes these therapies. The high frequency of changes to therapy in our cohort, in reaction to symptomatic complications and to patient preference, highlights the impact these decisions have on morbidity and quality of life. Our data describe coagulopathic risk factors and disproportionate risk of VTE in contrast to bleeding, which should be factored into individualized prescribing decisions.

Given the lack of definitive evidence, one key area to improve the quality of shared decision-making might be to champion greater patient involvement. This could include a consideration of oral or injectable therapy. The timeline for hematology and/or oncology outpatient review of anticoagulation plans should account for projected survival benefit. Outpatient review at four and ten weeks would miss 25% and 50% of the cohort, respectively, due to death. Early telephone consultation may be a high yield strategy, depending on the treatment goals.

### 4.5. Implications for Research

We plan to inform ongoing service provision in our trust with repeated service evaluations. Our study adds to the literature by describing adverse events with known risk factors relevant to clinical decision makers. Our rate of diagnostic imaging following stent insertion (55%) is similar to another retrospective review [10]. Many adverse events go un-investigated, and their clinical significance is difficult to ascertain. Interpreting conventional adverse outcome data and drawing conclusions about the success of anticoagulation strategies with regard to morbidity remains a challenge in this patient group.

Given patient preference for oral therapy, this should be considered prospectively in clinical trials. There is good evidence emerging to support the use of DOACs in treating and preventing cancer-associated thrombosis, yielding lower rates of recurrence, and their use is supported in guidelines [21]. The low rate of stent thrombosis when using antiplatelet agents is notable in the literature and is supported by a number of authors [6,15,17]. Many of our patients would be eligible for DOACs according to recent British Hematology Guidelines [22] for cancer-associated thrombosis, although evidence is lacking in this clinical setting. Bespoke prospective trials investigating the use of DOACs and/or anti-platelet therapy following stent insertion would help to assess the efficacy and safety of these treatment strategies. Validating a scoring system to stratify risk of VTE in groups of patients with mSVCO would help to better inform patient discussions on long-term therapy.

## 5. Conclusions

This retrospective analysis demonstrates that endovascular stenting for malignant superior vena cava obstruction achieves high technical success rates but occurs in a population with advanced disease and limited survival (median 2.4 months post stent insertion). We capture the evolving nature of anticoagulation management over time and across disciplines. Our findings reveal a significant risk of venous thromboembolism despite several anticoagulation strategies. The dynamic nature of anticoagulation management following stent insertion underscores the need for a systematic, multidisciplinary approach and deeper consideration of patient preference for what could be life-long therapy.

Comparisons between subgroups show the relative fitness of those with thrombotic complications. Many of the sicker patients following stent insertion remain un-investigated, and this is an important consideration for researchers and clinicians when interpreting adverse events. Future research could focus on prospective trials investigating direct oral anticoagulants and/or antiplatelet therapy in this specific clinical context, along with developing validated risk assessment tools to guide decision-making. Until stronger evidence emerges, we recommend early multidisciplinary evaluation of anticoagulation plans with scheduled reassessment tailored to the patient’s projected survival trajectory, ensuring that treatment remains aligned with the patient’s goals of care throughout their journey.

## Figures and Tables

**Figure 1 curroncol-32-00601-f001:**
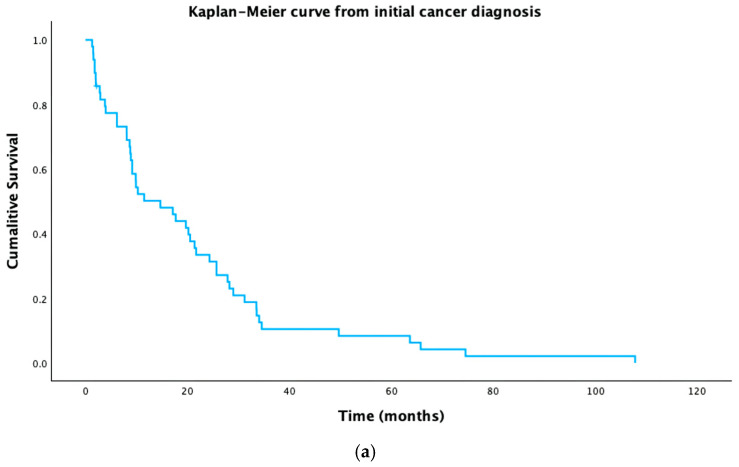
Kaplan–Meier curves of (**a**) patients from cancer diagnosis and (**b**) patients from stent insertion.

**Table 1 curroncol-32-00601-t001:** Baseline characteristics of all patients.

Descriptor	Number (Percentage/ Range)
**Total**	49 (100%)
Female sex	27 (55%)
Median age at diagnosis (years)	59 (IQR: 54–72)
**Malignancy classification**	
Lung cancer	40 (82%)
Non-small cell	23 (47%)
Adenocarcinoma	13 (47%)
- Squamous	8 (16%)
- Large cell neuro-endocrine	2 (4%)
- Not otherwise specified	5 (10%)
Small cell	9 (18.5%)
Mesothelioma	3 (6%)
Other malignancy	
- Breast	7 (14%)
- GI	1 (2%)
- Thyroid	1 (2%)
**Performance Status at diagnosis ^1^**	
0	14 (29%)
1	19 (39%)
2	7 (14%)
3	1 (2%)
Unknown	8 (16%)
**Metastatic disease status at time of stent insertion**	
No known metastatic disease outside of chest	13 (27%)
One site of metastatic disease outside of chest (nodal, visceral or bone).	24 (49%)
Two or more sites of metastatic disease outside of chest (nodal, visceral or bone).	12 (24%)
**Performance status at time of discharge following stent**	
0	1 (2%)
1	9 (18%)
2	12 (24%)
3	8 (16%)
Unknown	16 (33%)

^1^ Performance Status: European Cooperative Oncology Group (ECOG) performance status is a method to assess functional status and classify impairment ranked on scale of 0–4. PS: 0 indicates no functional impairment, 1 indicates a restriction in physically strenuous activity but an ability to carry out work of a light or sedentary nature and 3 indicates a capability of only limited selfcare, confined to either bed or chair for more than 50% of waking hours.

**Table 2 curroncol-32-00601-t002:** Anticoagulation therapy advised at discharge following stent insertion.

Agent	Number (Percentage/45)
Tx LMWH ^1^	28 (62%)
Px LMWH ^2^	5 (11%)
Nil	4 (9%)
Px LMWH and 75 mg Aspirin	3 (7%)
Tx LMWH and 75 mg Aspirin	1 (2%)
Apixaban 5 mg BD	1 (2%)
Apixaban 2.5 mg BD	1 (2%)
Dual anti-platelet therapy	1 (2%)
Aspirin monotherapy	1 (2%)

^1^ Tx LMWH: Therapeutically dosed low molecular weight heparin. ^2^ Px LMWH: Prophylactically dosed low molecular weight heparin.

## Data Availability

The data presented in this study are available on request from the corresponding author. The data are not publicly available due to privacy restrictions.

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
