# Peer review of "A Retrospective Analysis of Endovascular Stent Insertion for Malignant Superior Vena Cava Obstruction, Focusing on Anticoagulation Practices"

_curroncol, 2025, doi:10.3390/curroncol32110601_

Round 1

Reviewer 1 Report

Comments and Suggestions for Authors

Dear authors

you report a high rate of rethrombosis compared to ref.3

Your anticoagulation regimen was very inhomogeneous. Can you recognize any association between these parameters. There should be a table to demonstrate the relationship of the findings.

Author Response

Q1. Can you explain the re-stenosis rate higher than citation 3.

A1 Our re-stenosis rate of 4/49 is lower than the average reported by Fadgedet at al. (15%) in a large retrospective study of a patients with similar baseline characteristics to ours. The rate of stent thrombosis of our study was 7/49 (14%). Often thrombosis and recurrence are related; we did not perform histology or get secondary reviews of CT scans. I believe the difference in our data compared to the recurrence rates quoted in reference 3 relates to five key areas. The meta-analyses is pooling data from studies that analyse: 1) different types of SVCO (benign and malignant), 2) patients with different stages of oncological disease (earlier stages and more controllable) and 3) different types of stent (coated stents only). See below for further details. 4) All studies are hampered for accurate data given the rate of rapid decline following stent insertion, unknown whether due to recurrence. 5) Many studies included do not explicitly name an inclusion and exclusion criteria. We do not know how patients were selected. One of the advantages of our study is the near consecutive nature of all patients included with solid organ malignancy causing SVCO, and the characterisation of oncological characteristics. 

Further details:

When examining individual studies used by reference 3 (Aung et al) to determine recurrence rate at 6 months:

Different types of SVCO.

Qanadli’s review of 12 patients included patients with benign SVCS due to central venous catheters (80, post-radation fibrosis (2) and permanent pacemaker (1). Barshes et al., 2007 29% of patients was due to benign disease. These cohorts of patients present a significantly lower recurrence risk due to lack of progressive malignant  process.

Different stage of oncological disease.

The majority of our cohort present with late stage oncological disease, many have no further lines of systemic anti-cancer therapy treatments. This contrasts with the studies in the meta-analyses. Biedrager et al 2005, all patients included were chemo and radiation treatment naïve when the stents were inserted. 13/17 had chemotherapy or radiotherapy post stenting insertion, indicating their disease process is more modifiable and less likely to recur. 5 patients deteriorated within 3 months. We do not have an indication of the rate of CT imaging post stent insertion and so unable to define whether deterioration was caused by recurrence or alternative cause. Kim’s study (2003) 5/10 received chemo-radiation after stent insertion. Interestingly they introduced a dual anti-coagulation (warfarin) + anti-platelet (300mg ASA) approach after two instances of stent thrombosis in the ‘early part of the trial’. In Stock’s study, 10/14 patients had subsequent treatment with chemotherapy and radiotherapy. In this cohort 4/14 required re-intervention due to stent thrombosis. Kuo and colleagues (2017) select 12 patients with lung cancer to retrospectively analyse. They follow-up with interval scans at 3,6 and 12 months with a stent thrombosis rate of 17%. Patients with thrombosis at the time of stent insertion were treated with warfarin and those without clopidogrel. We do not know how they selected their patients. The title of the paper is ‘endovascular stenting for end-stage lung cancer with SVCS post first-line treatments’. They could therefore be at an earlier stage of treatment with more modifiable thrombotic risk factors and controllable disease; we do not know how fit and mobile the patients were. We hope to contribute to the literature by describing performance status and extent of metastatic disease to help qualify our patient cohort.

Different type of stent.  

Andersen and colleagues  (2015) present a cohort of patients more similar to ours, many of whom had exhausted many lines of chemotherapy and radiotherapy prior to stent insertion. They however assess outcomes in those receiving a coated nitinol stent, which is  known to reduce rates of thrombosis, possible recurrence due to thrombosis. They planned on routinely following up patients with CT imaging 1-3 months post insertion. Their mean follow-up time was 2.5 months (6/12 died in 1 month) and they don’t report how many people received a scan prior to dying.

Q2. ‘Your anticoagulation regimen was very inhomogeneous. Can you recognize any association between these parameters’

A2. The inhomogeneous drug regimens reflects the varied approach taken by different specialists with different priorities. The evidence for long-term anti-coagulation in the literature lacks definitive prospective trials leading to heterogenous practice. Our research reflects real-world practice at a tertiary teaching hospital. Not all patients were admitted under an oncologist and not all patients were discussed with a haematology specialist. Plans were initiated by IR, critical care, internal medical, oncology ad haematology specialists. In the absence of specific drug recommendations in IR, ESMO and haematology guidelines and there is a call in the paper for greater multi-disciplinary collaboration to better individualise care.

We felt on balance it would be unsatisfactory to draw associations between parameters due to the lower numbers in each treatment group and numerous confounding factors. The only group with decent numbers is those on treatment dose of low molecular weight heparin. We note in the complications section that 6/7 (86%) cases of radiologically proven stent-thromboses were prescribed a regimen of therapeutically dosed LMWH.  This might indicate a lack of efficacy of this therapy and as some authors suggest that anti-platelet therapy is essential for the prevention of early stent thrombosis (due to the speed of flow through the vein and platelet aggregation). We note in our discussion that low stent thrombosis rates in the literature include anti-platelet therapy. It might  however also indicate lack of compliance with injectable therapy. One other reviewer has asked about the outcomes for the 4 patients discharged without anticoagulation therapy; two patients returned with pulmonary embolism. We will include this in the discussion.

Actions in the paper: 

i) added to 4.2 

The rate of complications in our cohort, including recurrence and stent thrombosis was similar to some retrospective reviews, Azizi et al [4] found re-stenosis rates of 10.5% (95% CI 8.4%–12.6%), but higher than others [3]. Discrepancies are probably explained by pooled data capturing different patient groups, including those with benign SVCO and patients with earlier stages of cancer and more controllable disease. In contrast to previous studies, we present a wide variety of anticoagulation approaches. This reflects the real-world practice of prescribing decisions made by different specialists across two hospital sites with no definitive guideline recommendations on drug type and dose. Treatment groups were deemed too small to draw reliable associations between medication, effect and side-effects while adjusting for confounders.

ii)  added comment in 3.4.

Two out of four patients discharged without anticoagulation were admitted shortly afterwards with shortness of breath and later diagnosed with pulmonary embolism. Therapeutic anticoagulation was promptly commenced. The other two patients did not have interval imaging before deterioration.

References from above comments:

Reference 3: E. Y.-S. Aung et al., ‘Endovascular Stenting in Superior Vena Cava Syndrome: A Systematic Review and Meta-analysis’, Cardiovasc. Intervent. Radiol., vol. 45, no. 9, pp. 1236–1254, Sep. 2022, doi: 10.1007/s00270-022-03178-z.

Fagedet et al., ‘Endovascular Treatment of Malignant Superior Vena Cava Syndrome: Results and Predictive Factors of Clinical Efficacy’, Cardiovasc. Intervent. Radiol., vol. 36, no. 1, pp. 140–149, Feb. 2013, doi: 10.1007/s00270-011-0310-z.

Qanadli SD, El Hajjam M, Mignon F, de Kerviler E, Rocha P, Barré O, Chagnon S, Lacombe P. Subacute and chronic benign superior vena cava obstructions: endovascular treatment with self-expanding metallic stents. AJR Am J Roentgenol. 1999 Jul;173(1):159-64. doi: 10.2214/ajr.173.1.10397119. PMID: 10397119.

Barshes NR, Annambhotla S, El Sayed HF, Huynh TT, Kougias P, Dardik A, Lin PH. Percutaneous stenting of superior vena cava syndrome: treatment outcome in patients with benign and malignant etiology. Vascular. 2007 Sep-Oct;15(5):314-21. doi: 10.2310/6670.2007.00067. PMID: 17976332.

Bierdrager E, Lampmann LE, Lohle PN, Schoemaker CM, Schijen JH, Palmen FM, van der Heul C. Endovascular stenting in neoplastic superior vena cava syndrome prior to chemotherapy or radiotherapy. Neth J Med. 2005 Jan;63(1):20-3. PMID: 1571984

Kim YI, Kim KS, Ko YC, Park CM, Lim SC, Kim YC, Park KO, Yoon W, Kim YH, Kim JK, Ahn SJ. Endovascular stenting as a first choice for the palliation of superior vena cava syndrome. J Korean Med Sci. 2004 Aug;19(4):519-22. doi: 10.3346/jkms.2004.19.4.519. PMID: 15308841; PMCID: PMC2816884.

Stock KW, Jacob AL, Proske M, Bolliger CT, Rochlitz C, Steinbrich W. Treatment of malignant obstruction of the superior vena cava with the self-expanding Wallstent. Thorax. 1995 Nov;50(11):1151-6. doi: 10.1136/thx.50.11.1151. PMID: 8553270; PMCID: PMC475086.

Kuo TT, Chen PL, Shih CC, Chen IM. Endovascular stenting for end-stage lung cancer patients with superior vena cava syndrome post first-line treatments - A single-center experience and literature review. J Chin Med Assoc. 2017 Aug;80(8):482-486. doi: 10.1016/j.jcma.2017.04.005. Epub 2017 May 10. PMID: 28501315.

Andersen PE, Midtgaard A, Brenøe AS, Elle B, Duvnjak S. A new nitinol stent for use in superior vena cava syndrome. Initial clinical experience. J Cardiovasc Surg (Torino). 2015 Dec;56(6):877-81. Epub 2015 Jul 27. PMID: 26212865.

Reviewer 2 Report

Comments and Suggestions for Authors

The manuscript is written on twelve pages, two of which contain references. It includes two figures, one video and three tables. The manuscript is written in the standard English language and uses sophisticated scientific methods.

Content suggestions:

  1. Was anti-Xa activity of LMWH and antithrombin level measured in any of the included

patients ?    

  1. According to the Table 2, there was no anticoagulant treatment advised at discharge

            following stent insertion advised in 4 patients. Can the Authors state whether

            rethrombosis occurred in them or any details about their outcome ?

Author Response

Q1. Anti-XA activity and anti-thrombin level

A1. This was not routinely measured in patients and the results were not reliably accessible to researchers. This would be an interesting insight given the recurrence rate. I note in the 2024 British Haematology Guidelines for Cancer Associated thrombosis that the two studies (Carrier et al., 2009, Inhaddadene et al., 2014) retrospectively assessing recurrent thromboembolism in cancer patients failed to measure anti-Xa and antithrombin levels. The conclusion of the BSH is that there is no evidence that escalating doses of ‘LMWH to supratherapeutic doses is associated with a better outcome’. This remains however standard practice. 

References:

Carrier M, Le Gal G, Cho R, Tierney S, Rodger M, Lee AY. Dose escalation of low molecular weight heparin to manage recurrent venous thromboembolic events despite systemic anticoagulation in cancer patients. J Thromb Haemost. 2009; 7(5): 760–765.

Ihaddadene R, Le Gal G, Delluc A, Carrier M. Dose escalation of low molecular weight heparin in patients with recurrent cancer-associated thrombosis. Thromb Res. 2014; 134(1): 93–95.

Q2. No anticoagulant treatment advised for 4 patients. Can the authors confirm the re-thrombosis rate?

A2. I can confirm that 2 patients were promptly admitted with shortness of breath and diagnosed with pulmonary embolism and commenced on anti-coagulation. The two other patients deteriorated with no repeat imaging. All four patients were admitted under medical team rather than oncologists.

Actions in the paper in response to comments: 

i) added to 4.2 

The rate of complications in our cohort, including recurrence and stent thrombosis was similar to some retrospective reviews, Azizi et al [4] found re-stenosis rates of 10.5% (95% CI 8.4%–12.6%), but higher than others [3]. Discrepancies are probably explained by pooled data capturing different patient groups, including those with benign SVCO and patients with earlier stages of cancer and more controllable disease. In contrast to previous studies, we present a wide variety of anticoagulation approaches. This reflects the real-world practice of prescribing decisions made by different specialists across two hospital sites with no definitive guideline recommendations on drug type and dose. Treatment groups were deemed too small to draw reliable associations between medication, effect and side-effects while adjusting for confounders.

ii)  added comment in 3.4.

Two out of four patients discharged without anticoagulation were admitted shortly afterwards with shortness of breath and later diagnosed with pulmonary embolism. Therapeutic anticoagulation was promptly commenced. The other two patients did not have interval imaging before deterioration.

Reviewer 3 Report

Comments and Suggestions for Authors

The article is very well written and addresses an uncommon but important topic in the field. The field involves: Oncology, Vascular Surgery, Interventional Radiology. 

These patients are usually quite sick and symptomatic and the use of anticoagulants plays a key role in their management.

Would be very difficult to conduct prospective randomized trials to determine the "best" option so this contribution is very valuable.

Author Response

he article is very well written and addresses an uncommon but important topic in the field. The field involves: Oncology, Vascular Surgery, Interventional Radiology. 

These patients are usually quite sick and symptomatic and the use of anticoagulants plays a key role in their management.

Would be very difficult to conduct prospective randomized trials to determine the "best" option so this contribution is very valuable.

A:

Many thanks for these encouraging comments. We would agree with the above, the cohort are sick, highly medicalised and follow-up is challenging. The care lives on the boundaries of many specialist and we believe it is important to bring the disciplines together. Thank you for your support. 

Actions in the paper in response to reviewers comments: 

i) added to 4.2 

The rate of complications in our cohort, including recurrence and stent thrombosis was similar to some retrospective reviews, Azizi et al [4] found re-stenosis rates of 10.5% (95% CI 8.4%–12.6%), but higher than others [3]. Discrepancies are probably explained by pooled data capturing different patient groups, including those with benign SVCO and patients with earlier stages of cancer and more controllable disease. In contrast to previous studies, we present a wide variety of anticoagulation approaches. This reflects the real-world practice of prescribing decisions made by different specialists across two hospital sites with no definitive guideline recommendations on drug type and dose. Treatment groups were deemed too small to draw reliable associations between medication, effect and side-effects while adjusting for confounders.

ii)  added comment in 3.4.

Two out of four patients discharged without anticoagulation were admitted shortly afterwards with shortness of breath and later diagnosed with pulmonary embolism. Therapeutic anticoagulation was promptly commenced. The other two patients did not have interval imaging before deterioration.

Round 2

Reviewer 1 Report

Comments and Suggestions for Authors

Dear authors

thank you for the clarifications.

Due to the very inhomogeneous data, there should be a table including all 49 pt. and their results after stenting including anticoagulation and complications,

Author Response

Comment 1: 

Thank you for the clarifications. Due to the very inhomogeneous data, there should be a table including all 49 pt. and their results after stenting including anticoagulation and complications.

Response 1: 

Please find attached a table of all 49 patients. Due to the table size, the file would be considered best found in Appendix A. however this is open to advice from the editors. The table details age (given as range due to risk of identification), cancer type (lung and mesothelioma grouped together), type of stent, anticoagulation, complication, treatment change and survival. 

Further edits have been made, and to be found in new revised copy of manuscript):

Edits: I have referred to the new table in section 3 Results.

Edits: In section 3.3 I have updated Table 2 on anticoagulation regimens for clarity in concordance with the new supplementary table.  

Edits: In section 3.4 I have clarified and changed the median survival for stent thrombosis specifically rather than grouping with recurrence, stenosis and thrombosis together. This point was also clarified in section 4.1 and 4.4 of the discussion. This was done because it's more relevant to discussions specifically regarding anti-coagulation.

Reviewer 2 Report

Comments and Suggestions for Authors

I would like to tank the Authors for the revision and resubmission of the manuscript. Currently, the text is clearer and provides sufficient information about the anticoagulation therapy after endovascular stent insertion for malignant superior vena cava obstruction. I sincerely appreciate their effort to provide novel data in this field of study and I recommend the publication of this version of the paper. 

Author Response

Comments: Thank you for your comments and recommendations.